# Enhancing Large Language Models with Neurosymbolic Reasoning for Multilingual Tasks

**Sina Bagheri Nezhad**                                    SINA.BAGHERINEZHAD@PDX.EDU
**Ameeta Agrawal**                                        AMEETA@PDX.EDU
*Portland State University, USA*

**Editors:** Leilani H. Gilpin, Eleonora Giunchiglia, Pascal Hitzler, and Emile van Krieken

## Abstract

Large language models (LLMs) often struggle to perform multi-target reasoning in long-context scenarios where relevant information is scattered across extensive documents. To address this challenge, we introduce NeuroSymbolic Augmented Reasoning (NSAR), which combines the benefits of neural and symbolic reasoning during inference. NSAR explicitly extracts symbolic facts from text and generates executable Python code to handle complex reasoning steps. Through extensive experiments across seven languages and diverse context lengths, we demonstrate that NSAR significantly outperforms both a vanilla RAG baseline and advanced prompting strategies in accurately identifying and synthesizing multiple pieces of information. Our results highlight the effectiveness of combining explicit symbolic operations with neural inference for robust, interpretable, and scalable reasoning in multilingual settings.

## 1. Introduction

Large language models (LLMs) have achieved impressive progress in natural language processing, yet their ability to handle *long-context, cross-lingual reasoning* remains limited (Agrawal et al., 2024). In real-world multilingual scenarios, information is often scattered across languages and embedded within lengthy documents, complicating retrieval and coherent reasoning. This challenge is exacerbated by the *lost-in-the-middle* phenomenon, where LLMs struggle to retain and integrate relevant information from extended contexts (Liu et al., 2023). Although retrieval-augmented generation (RAG) frameworks help narrow the context window, LLMs still frequently fail to combine dispersed facts into a consistent chain of reasoning, resulting in hallucinations and inconsistencies (Zhu et al., 2024; Jiang et al., 2024).

In this work, we introduce **NeuroSymbolic Augmented Reasoning (NSAR)**—a neurosymbolic method that explicitly merges symbolic reasoning with neural inference. NSAR centers on a *neurosymbolic prompt* that instructs LLMs to extract structured symbolic facts and generate executable code, thus offering a verifiable and interpretable reasoning pathway for complex tasks. By integrating symbolic logic into a retrieval-based framework, NSAR improves interpretability, consistency, and accuracy when working with multilingual and long-context data.

The main contributions of our work are as follows:

- We propose NSAR, a neurosymbolic reasoning framework that couples retrieval with explicit symbolic reasoning. We develop a dedicated NSAR prompt to structure information into symbolic representations and guide the model in generating executable Python code for final verification.

- We show that combining *Chain of Thought (CoT), ReAct, and Self-Reflection* prompting strategies with neurosymbolic reasoning leads to further performance gains.

- Through extensive experiments on cross-lingual, long-context tasks, we demonstrate that NSAR substantially outperforms purely retrieval-based and neural-only methods.

## 2. Related Works

**Long-Context and Cross-Lingual Challenges**  Large language models (LLMs) frequently exhibit a *lost-in-the-middle* phenomenon in extended contexts (Liu et al., 2023; Xu et al., 2024), causing them to overlook crucial information. Retrieval-augmented frameworks such as LONGEMBED (Zhu et al., 2024), LongRAG (Jiang et al., 2024) and DR-RAG (Hei et al., 2024) alleviate this issue but mostly target monolingual settings. McCrolin (Limkonchotiwat et al., 2024) extends these ideas to cross-lingual retrieval, yet it remains computationally intensive. Bridging queries and contexts in different languages is still difficult (Li et al., 2024; Hengle et al., 2024). Methods such as OPTICAL (Huang et al., 2023) and XAMPLER (Lin et al., 2024) improve multilingual retrieval but face scalability limitations in very long contexts. Parameter-efficient solutions, such as Sparse Fine-Tuning Masks (SFTMs) (Litschko et al., 2022) and adapters, achieve strong multilingual retrieval while reducing resource overhead, but they do not fully address multi-target reasoning.

**Advanced Prompting Strategies**  Prompting methods like Chain-of-Thought (CoT) (Wei et al., 2022), ReAct (Yao et al., 2023b), and Self-Reflection (Renze and Guven, 2024) improve LLM reasoning by encouraging more explicit intermediate steps. The Tree-of-Thoughts (ToT) approach (Yao et al., 2023a) further expands these pathways, enhancing structured reasoning. However, such methods rely on text-based explanations that may lack verifiability.

**Neurosymbolic Reasoning**  Neurosymbolic approaches offer interpretability and robustness by coupling neural networks with symbolic logic. Hybrid systems like LINC (Olausson et al., 2023) and PAL (Gao et al., 2023) produce executable code, improving transparency and accuracy. Similarly, incorporating symbolic modules into decision-making (Fang et al., 2024) enhances reliability.

## 3. System Architecture and Approach

Our system, as visualized in Figure 1, is composed of two distinct layers. First, a *Retrieval Component* efficiently narrows down the long, multilingual context using a two-phase Retrieval-Augmented Generation (RAG) framework. Then, the *NeuroSymbolic Augmented Reasoning (NSAR)* component builds upon this retrieved context to perform explicit neurosymbolic reasoning. NSAR employs a specialized prompt that instructs the language model to extract structured symbolic facts and generate executable Python code for deterministic reasoning.

### 3.1. Retrieval Component

The retrieval component extracts the most relevant chunks of information from extensive, multilingual contexts using a Retrieval-Augmented Generation framework.

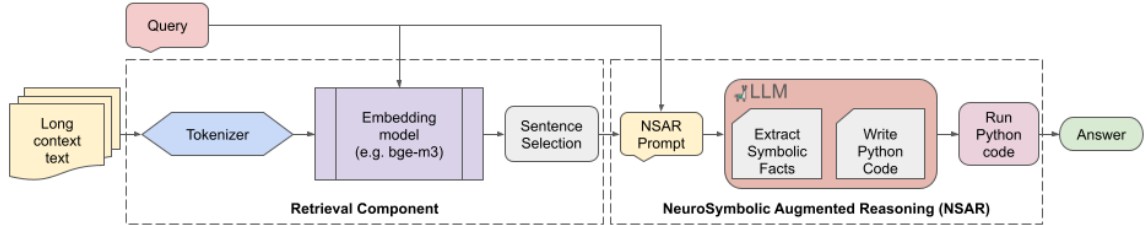

Figure 1: High-level overview of the system, illustrating the two layers: the *retrieval component* (left) and the *neurosymbolic reasoning component* (NSAR, right). First, the long context and query are tokenized and embedded to select the most relevant sentences. Next, the NSAR prompt directs the LLM to extract symbolic facts and generate Python code, which is then executed to produce the final answer.

**Tokenization and Embedding** The input context, which may comprise hundreds of thousands of words across multiple languages, is first segmented into sentences and embedded to capture semantic nuances. We use the Punkt tokenizer (Kiss and Strunk, 2006) for segmentation and the multilingual *bge-m3* model (Chen et al., 2024) to generate sentence embeddings in 1024-dimensional embedding size.

**Candidate Selection and Model Input** Next, we compute the semantic distance between each sentence embedding and the query, selecting the top $k$ most relevant sentences based on a tunable hyperparameter (with $k$ values of 3, 5, 10, 20, and 50 evaluated in our experiments). These sentences are then passed to the language model as input for the subsequent neurosymbolic reasoning process.

### 3.2. NeuroSymbolic Augmented Reasoning (NSAR)

Purely neural approaches to long-context question answering often struggle with reliability, interpretability, and logically integrating multiple pieces of information. While recent prompting techniques (e.g., Chain-of-Thought, ReAct, and Self-Reflection) have improved reasoning, they remain reliant on implicit neural processes, leaving little room for explicit verification or modular correction. To address these limitations, we introduce the NeuroSymbolic Augmented Reasoning (NSAR) component which integrates structured symbolic representations within the neural architecture by extracting symbolic facts and generating executable Python code for reasoning. This approach bridges the gap between the *flexibility and fluency* of LLMs and the *interpretability and rigor* of symbolic methods. Neural systems excel at language understanding and generation, but often fail in complex scenarios requiring multiple reasoning steps, such as comparing multiple facts, deducing the "largest" or "smallest" value, or verifying constraints across scattered pieces of information. Symbolic methods, by contrast, excel in structured reasoning but can be brittle when parsing unstructured text. By coupling a language model with a symbolic layer, NSAR enhances interpretability by providing an explicit record of extracted facts and logical steps, enabling users to audit and verify the model's reasoning. Additionally, it improves reliability by reducing errors in compositional tasks by systematically comparing and fusing pieces of information through symbolic code execution.

We design an *NSAR prompt* to guide the reasoning process through three distinct stages:

**1. Symbolic Fact Extraction** First, the model is instructed to identify all relevant facts in the provided context and represent them in a structured, symbolic format. For instance, if the context contains lines such as *"The special magic Cairo number is: 1234567"* and *"The special magic Mumbai number is: 9999999"*, the model generates:

```
FACT("Cairo", "special_magic_number", 1234567)
FACT("Mumbai", "special_magic_number", 9999999)
```

**2. Python Code Generation** Next, the model is prompted to produce concise, executable Python code that uses the extracted symbolic facts to answer the question. Instead of implicitly inferring logic through text, the Python code can contain explicit comparisons (`>`, `<`, `==`), data structures (lists, dictionaries), or domain-specific libraries. In the case of identifying the *largest special magic number*, this code might look like:

```
numbers = [1234567, 9999999]
answer = max(numbers)
```

The logic here can be arbitrarily extended to handle more complex reasoning steps (e.g., filtering facts, applying constraints, or computing aggregates). Once the LLM generates the Python code, it is executed in a controlled environment.

**3. Final Answer Extraction** Finally, the answer is determined by executing the generated Python code. This guarantees a concise, verified response and prevents any contradictory or incoherent rationales that might arise from purely text-based reasoning. In other words, while the LLM might propose a final textual answer, the *actual* answer delivered to the user is the deterministic output of the code execution.

As such, NSAR prompt structure ensures that the language model provides an interpretable chain of reasoning, resulting in a Python snippet that can be independently executed and verified. The template of NSAR prompt is shown below:

---

**NSAR Prompt Template**

You are a helpful assistant that employs a neurosymbolic method. Given the following context and question, please follow these steps:
1. Extract all relevant facts from the context and represent them as symbolic facts using the format `FACT(entity, attribute, value)`.
2. Generate executable Python code that uses the extracted symbolic facts to compute the final answer.
3. Finally, output only the final answer.
#CONTEXT
{text}
#ENDCONTEXT
#QUESTION
What is the largest special magic number?

---

In this work, "neurosymbolic" denotes the hybrid of explicit fact extraction with deterministic code execution. While our current triples support only simple attribute logic, extending to richer formalisms (e.g. first-order rules or constraint solvers) would more fully realize the neurosymbolic ideal.

## 4. Experiments

### 4.1. Dataset

We evaluate the proposed approach on a question answering task encompassing long contexts and multiple languages. We adopt and extend the mLongRR dataset (Agrawal et al., 2024) by increasing the maximum context length to 512,000 words and expanding the language set to English, Vietnamese, Swahili, Persian, Russian, Hindi, and Arabic. This setting incorporates multiple scripts (e.g., Cyrillic, Devanagari) and introduces a cross-lingual challenge: the query is always in English while the context (haystack) is in one of the seven languages. In addition, we randomly place three "needles" (target sentences) within each context haystack, requiring the model to retrieve and compare multiple pieces of relevant information.

The contexts consist of news articles ranging in size from 2k, 8k, 16k, 32k, 64k, 128k, 256k, and 512k words. Embedded within these contexts are needles such as: "The special magic {city} number is: {number}", where {city} is randomly chosen from 23 city names translated into all seven target languages, and {number} is a random 7-digit number (Agrawal et al., 2024; Team, 2024; Anthropic, 2024). Needles appear in the same language as the haystack, ensuring a consistent linguistic setting. Additional details on the needle translation and city selection process can be found in Appendix A.1.

### 4.2. Evaluation

To assess the effectiveness of NSAR in handling complex, cross-lingual reasoning tasks, we adopt an evaluation protocol based on a multi-target scenario—the 3-needles test. Previous studies have reported that the 3-needle scenario is particularly challenging for LLMs operating in long-context settings and low-resource languages (Agrawal et al., 2024). In this test, three needles are randomly placed throughout the context, and the model is prompted with the query: "What is the largest special magic number?" This setup requires the model to locate and compare multiple pieces of information to produce the correct answer, reflecting real-world scenarios where critical information is scattered.

We run experiments across eight different context lengths (2k, 8k, 16k, 32k, 64k, 128k, 256k, and 512k words) and five values of $k$ for retrieved sentences (3, 5, 10, 20, and 50). For each setting, the three needles are randomly positioned in the text. We use accuracy as our main metric, defined as the percentage of cases in which the model correctly identifies the correct response by effectively integrating multiple retrieved facts. The results are reported in terms of average accuracy computed over two runs.

### 4.3. Models and Baseline Methods

We evaluate our approach using two large language models. GPT-4o-mini is a smaller, resource-efficient variant of GPT-4o (OpenAI et al., 2024), designed to trade off some generative capacity for faster inference and lower memory usage, and Llama 3.2 (90B) which is an expanded successor to Llama 2 (Touvron et al., 2023), with refined training procedures and a broader pretraining corpus.

We include three relevant baseline prompting strategies like Chain-of-Thought (Wei et al., 2022), ReAct (Yao et al., 2023b), and Self-Reflection (Renze and Guven, 2024) in our evaluation which have demonstrated success in boosting reasoning performance.

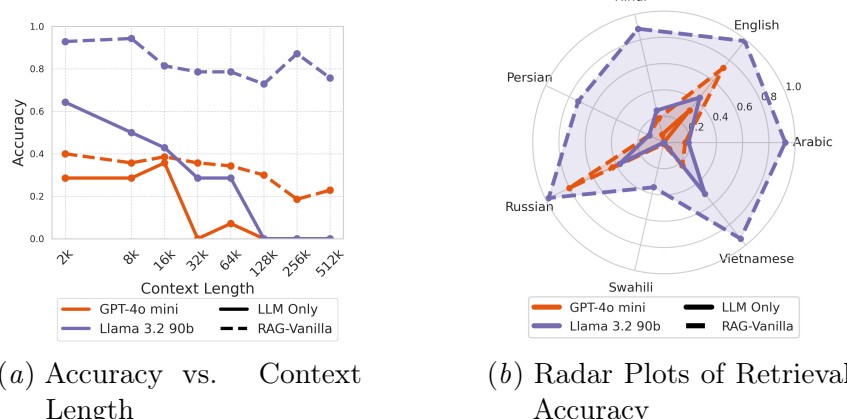

(a) Accuracy vs. Context Length

(b) Radar Plots of Retrieval Accuracy

Figure 2: Accuracy as a function of context length and across seven context languages.

## 5. Results and Analysis

### 5.1. LLM-Only *vs.* RAG-Vanilla

We compare an LLM-only approach, where the entire context is fed directly to the model, with a RAG-Vanilla approach that employs the retrieval component to narrow the context (by identifying the top-$k$ most relevant sentences) before further processing with the LLM. This selective context narrowing not only improves retrieval accuracy but also significantly reduces computational costs by minimizing the total token count processed by the model. In both methods, we use the same prompt adapted from previous work (see Appendix A.2). The performance results for the RAG-Vanilla approach shown in the figures represent the average accuracy across all k-values and evaluated languages.

**Context length**   From the results presented in Figure 2(a), we observe that both GPT-4o-mini and Llama 3.2 exhibit declining accuracy as context length increases in LLM only (solid lines) scenario, with accuracy dropping to almost 0% as context lengths reach 128K words. In contrast, narrowing the input to a small set of top-relevant sentences using RAG-Vanilla (dashed lines) enables these models to maintain high accuracy even in long-context scenarios (over half a million words).

**Accuracy by context language**   Figure 2(b) shows radar plots comparing accuracy across seven context languages using English prompts for the 3-Needles test. We observe that RAG-Vanilla consistently outperforms the LLM-only baseline across all tested languages, including those with non-Latin scripts such as Persian, Russian, Hindi, and Arabic. Notably, even languages like Swahili and Vietnamese, which differ significantly from English in morphology, benefit from the narrowed context, highlighting the method's cross-lingual robustness.

**Error analysis**   Although the retrieval-based approach demonstrates significant improvements in accuracy, a closer examination of incorrect cases reveals important insights into its limitations. We analyze the error rates, distinguishing between errors occurring during the embedding retrieval stage and those during the language model's response generation. The errors are categorized into two types:

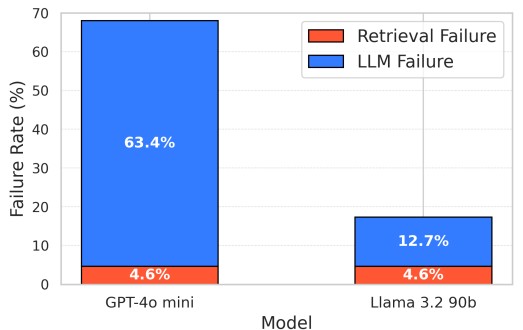

(a) Retrieval failures (red) versus LLM failures (blue) for GPT-4o-mini and Llama 3.2.

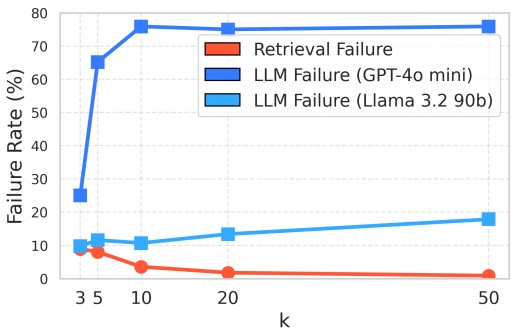

(b) Retrieval errors and LLM errors as $k$ (the number of retrieved sentences) increases.

Figure 3: A combined view of errors in RAG-Vanilla (failure rate % of total queries).

- Retrieval error: The correct sentence is not included among the top-$k$ retrieved segments, indicating that the embedding model failed to identify the relevant content.

- LLM error: The language model fails to extract or correctly reason about the target information, even when the relevant sentence is retrieved.

Error rates for each category are calculated as the percentage of total queries that result in that specific type of failure.

Figure 3(a) shows the distribution of retrieval errors (red) versus LLM errors (blue) for GPT-4o-mini and Llama 3.2 90b using RAG-Vanilla. Although retrieval errors remain relatively low (4.6% for both models), the LLM error rates are quite substantial for GPT-4o-mini (63.4%) and Llama 3.2 90b (12.7%), indicating that while RAG-Vanilla effectively narrows the context, the language model itself continues to struggle with multi-target reasoning. We also evaluated how increasing $k$ (the number of retrieved sentences) influences retrieval errors, as illustrated in Figure 3(b). Although a larger $k$ reduces the likelihood of missing relevant information, it introduces additional distractors that can complicate the model's reasoning process.

Based on these results, we identify several key factors contributing to LLM failures:

- Increased distractors: Multiple similar sentences can confuse the model's reasoning process.

- Inconsistent answer prioritization: Determining which label is "largest" or "most relevant" can be nontrivial when multiple valid options exist.

- Ambiguity in sentence ranking: Even with successful retrieval, the model may incorrectly prioritize semantically similar sentences when generating its final response.

These findings underscore the need for more advanced methods—such as neurosymbolic approaches—to improve multi-target reasoning beyond the gains offered by simple context narrowing.

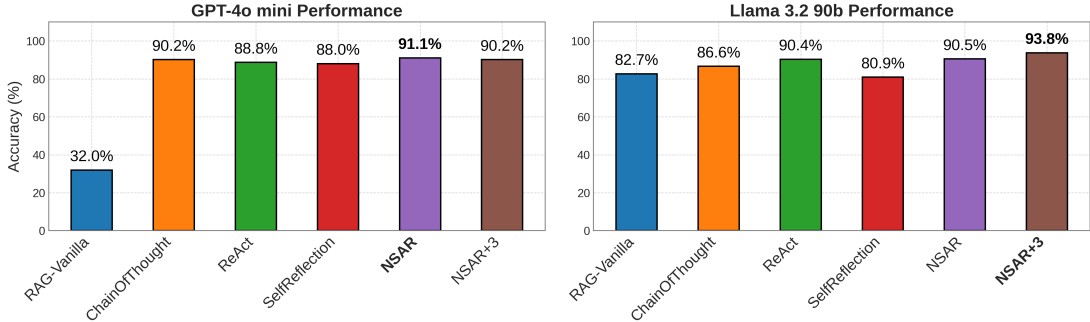

Figure 4: Overall accuracy of GPT-4o-mini (left) and Llama 3.2 90b (right) under different reasoning strategies (RAG-Vanilla, CoT, ReAct, Self-Reflection, NSAR, and a combined approach which combines NSAR with other reasoning methods (*NSAR+3*)).

## 5.2. NeuroSymbolic Reasoning (NSAR)

Although RAG narrows down the context and alleviates many challenges inherent to long input passages, it alone does not guarantee robust multi-target reasoning. To address this shortcoming, we enhance our baseline RAG system with NSAR component.

As baselines, besides *RAG-Vanilla*, we evaluate three prompting-based methods—*Chain-of-Thought* (CoT), *ReAct*, and *Self-Reflection*. We also experiment with a hybrid approach (*NSAR+3*) that combines NSAR with all three prompting strategies (Chain-of-Thought, ReAct, Self-Reflection, NSAR, NSAR+3).

As shown in Figure 4, the *RAG-Vanilla* baseline lags behind in multi-target reasoning, confirming that narrowing the context alone does not suffice to fuse and compare multiple pieces of information. In contrast, our proposed approach NSAR substantially improves accuracy by leveraging explicit symbolic extraction and Python-based reasoning. Moreover, combining NSAR with CoT, ReAct, and Self-Reflection (*NSAR+3*) yields the highest accuracy overall. Notably, for GPT-4o-mini, *NSAR* achieves 91.1%, followed closely by *NSAR+3* and *CoT*, both at 90.2%, whereas for Llama 3.2, *NSAR+3* attains the highest performance at 93.8%. These findings suggest that integrating explicit symbolic reasoning can fill critical gaps in retrieval-augmented generation, particularly for complex tasks that demand robust compositional inference.

**Accuracy by context language** Figures 5(*a*) and 5(*b*) provide a more granular perspective on how each approach performs across seven context languages. Each cell represents the accuracy (%) of a particular approach–language pair, revealing where certain strategies excel or fall short.

Across both heatmaps, *NSAR* and *NSAR+3* maintain high accuracy in most languages, confirming the benefits of explicit symbolic reasoning for multi-target retrieval tasks. In contrast, baseline methods (*RAG-Vanilla*) and single-step prompting strategies (Chain-of-Thought, ReAct, Self-Reflection) exhibit more variability and struggle in specific languages. Notably, languages such as Swahili and Arabic appear more challenging, yet neurosymbolic approaches still achieve competitive performance. These language-specific patterns underscore the importance of robust, compositional reasoning—particularly in cross-lingual or lower-resource settings.

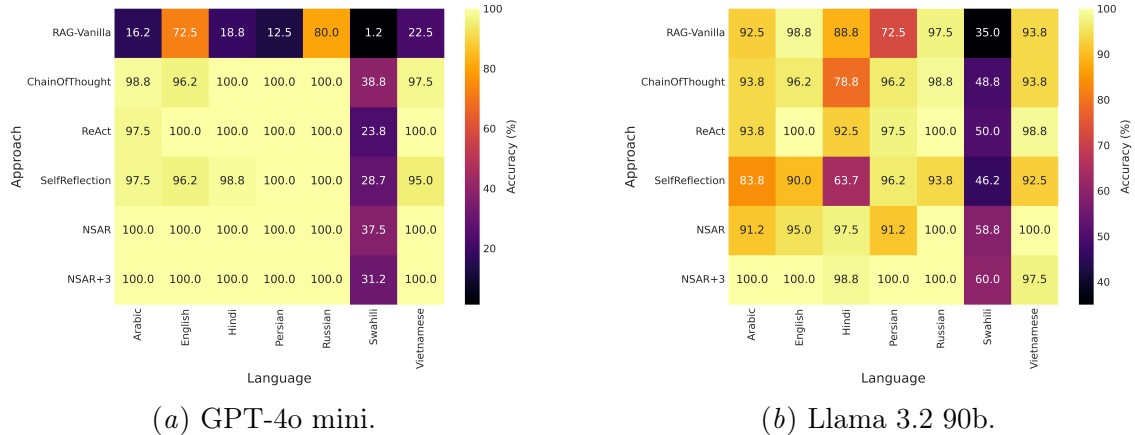

(a) GPT-4o mini.    (b) Llama 3.2 90b.

Figure 5: Heatmaps illustrating the accuracy (%) of different approaches (rows) across seven context languages (columns). Darker cells indicate **lower** accuracy, while lighter cells indicate **higher** accuracy.

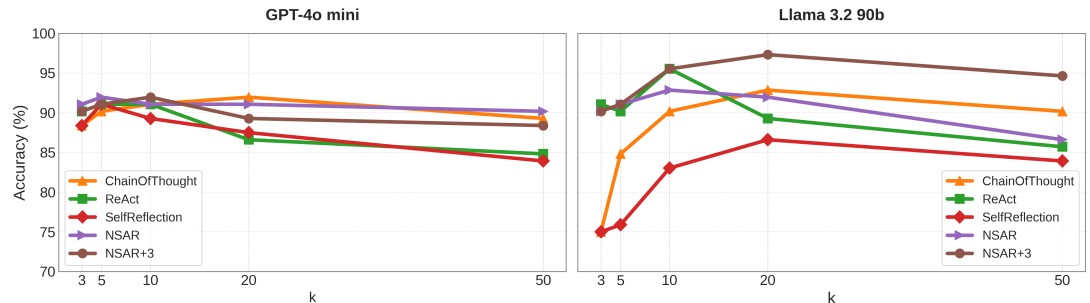

Figure 6: Accuracy versus $k$ (the number of retrieved sentences) for GPT-4o-mini (left) and Llama 3.2 (right).

**Effect of $k$ on performance**    Figure 6 shows how different reasoning strategies perform as we vary the number of retrieved sentences $k$ (3, 5, 10, 20, and 50) for GPT-4o-mini (left) and Llama 3.2 90b(right). Several trends are evident. First, at low values of $k$, all methods tend to have lower accuracy, likely due to the increased chance of missing key information during retrieval. As $k$ increases, accuracy generally improves up to a point. However, very large $k$ (e.g., 50) can introduce additional distractors, leading to a decline in performance. This aligns with our earlier observations that, while a broader retrieval scope reduces the risk of overlooking relevant facts, it can also complicate the model's reasoning by introducing more non-essential content.

Comparing across models, Llama 3.2 shows more pronounced fluctuations as $k$ increases, suggesting it is more sensitive to context size and potential distractors. In contrast, GPT-4o-mini maintains relatively stable performance at intermediate $k$ values. Notably, NSAR+3 consistently outperforms purely neural prompting methods in Llama 3.2, whereas GPT-4o-mini exhibits closer competition among NSAR and Chain-of-Thought. Overall, these findings highlight the importance of carefully tuning $k$ to balance retrieval breadth and

processing load, while also demonstrating that neurosymbolic reasoning can mitigate many of the challenges introduced by larger context windows.

**Error analysis for NSAR and NSAR+3**   Figure 7 illustrates the distribution of error types for GPT-4o-mini and Llama 3.2 under the *NSAR* and *NSAR+3* methods, categorizing failures into two types:

- Facts: The model retrieved the correct segments but failed to extract the target fact from the input.

- Code: Although the target fact was extracted correctly, the model produced incorrect or incomplete Python code, leading to an erroneous final answer.

The distribution of fact-extraction versus code-generation failures varies notably between the two models and across the two neurosymbolic methods. In *NSAR* for GPT-4o-mini, most errors stem from fact extraction, whereas Llama 3.2 primarily struggles with code generation. This suggests that GPT-4o-mini's neurosymbolic pipeline struggles in locating the correct sentences to extract facts, while Llama 3.2 successfully extracts facts but sometimes produces flawed Python code. Moving from *NSAR* to *NSAR+3* reverses this trend for Llama 3.2, significantly reducing code-generation failures but leading to more fact-extraction issues. Meanwhile, GPT-4o-mini exhibits a small increase in fact-extraction errors and a modest rise in code-generation errors when switching to *NSAR+3*.

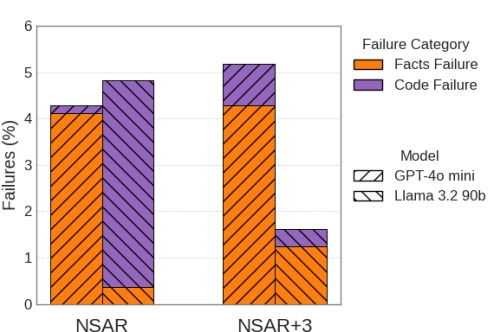

Figure 7: Failure rate (% of total queries) for fact extraction vs. code generation errors under *NSAR* and *NSAR+3*.

Overall, these results imply that while neurosymbolic reasoning substantially mitigates retrieval shortcomings, the balance between accurate fact extraction and correct code generation can shift depending on the underlying model and the specific prompting strategy.

## 6. Conclusion

We presented a neurosymbolic reasoning module (NSAR) to address the challenges of multi-target reasoning in long-context, cross-lingual settings. While RAG effectively narrows the context and reduces computational overhead, purely neural models often struggle to integrate and compare multiple pieces of information. NSAR overcomes these limitations by extracting symbolic facts and generating executable Python code, enabling transparent, verifiable reasoning. Our experiments across seven languages demonstrate that NSAR significantly outperforms both the vanilla RAG baseline and advanced prompting strategies (Chain-of-Thought, ReAct, and Self-Reflection). Future work will expand NSAR's symbolic scope beyond numeric lookups—into graph queries, set operations, and constraint satisfaction—and evaluate on broader tasks to test generalizability of our reasoning pipeline.

## Acknowledgments

We thank the anonymous reviewers for their valuable feedback.

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

## Appendix A. Prompt and Needle Templates

This appendix presents the templates and data used in our experiments. For the corresponding versions of Needles in other languages (Swahili, Vietnamese, Persian, Hindi, Arabic, and Russian), please refer to the anonymous repository [1].

### A.1. Needle Template

`"The special magic {city} number is {number}."`

Here, `number` represents a randomly generated 7-digit number, and `city` is selected at random from the list below (with the city names translated into the context language):

- Chicago
- Yangon
- Antananarivo
- Colombo
- Almaty
- Sydney
- Mexico City
- Seattle
- Lagos
- Amsterdam
- Belgrade
- Cairo
- Baghdad
- Damascus
- Kigali
- Dakar
- Sofia
- Victoria
- Tashkent
- Mumbai
- Barcelona
- Amman
- Toronto

### A.2. Prompt Templates

This appendix presents the complete set of prompt templates used in our experiments, each designed to elicit concise and direct answers from a large language model (LLM). All prompts share the convention that contextual information is enclosed between `#CONTEXT` and `#ENDCONTEXT`, followed by a specific question.

---

1. https://anonymous.4open.science/r/prompts-template-D729

- **Vanilla Prompts**: Adapted from Agrawal et al. (2024), provide a baseline retrieval-focused setup without any specialized reasoning strategy.

- **Chain-of-Thought (CoT)**: Encourages the LLM to articulate its reasoning steps in natural language before arriving at a final answer, aiming to improve transparency and correctness.

- **ReAct**: Combines "thinking" (Thought) and "acting" (Action) phases, prompting the LLM to explicitly delineate its internal reasoning and the subsequent steps taken to reach an answer.

- **Self-Reflection**: Prompts the model to not only provide a step-by-step solution but also critically review its reasoning for potential errors, thus refining the final answer.

- **NSAR**: Introduces a neurosymbolic prompt that extracts symbolic facts and generates executable Python code to produce a deterministic, verifiable answer.

- **NSAR+3**: A hybrid method that combines neurosymbolic reasoning with Chain-of-Thought, ReAct, and Self-Reflection, offering a comprehensive approach to both interpretability and robust logic.

Below, we detail each prompt template:

---

**Vanilla Prompt Template**

You are a helpful AI bot that answers questions for a user. Keep your response short and direct.
#CONTEXT
{text}
#ENDCONTEXT
#QUESTION
What is the largest special magic number? Don't give information outside the document or repeat your findings. If the information is not available in the context, respond UNANSWERABLE.

---

**Chain-of-Thought Prompt Template**

You are a helpful assistant. Below is a context and a question. Please think through the problem step by step, detailing your reasoning process thoroughly before providing your final answer.
#CONTEXT
{text}
#ENDCONTEXT
#QUESTION
What is the largest special magic number? Please explain your reasoning process step by step before providing the final answer.

---

---

**ReAct Prompt Template**

You are a helpful assistant. Below is a context and a question. For this question, please follow these steps:
1. Provide your thought process, prefixed with "Thought:".
2. Describe the action you would take, prefixed with "Action:".
3. Finally, state the final answer, prefixed with "Final Answer:".
#CONTEXT
{text}
#ENDCONTEXT
#QUESTION
What is the largest special magic number?

---

**Self-Reflection Prompt Template**

You are a helpful assistant. Below is a context and a question. For the given question, please:
1. Provide a detailed, step-by-step explanation of your reasoning.
2. Critically review your reasoning to ensure it is sound.
3. Finally, state your final answer.
#CONTEXT
{text}
#ENDCONTEXT
#QUESTION
What is the largest special magic number?

---

**NSAR Prompt Template**

You are a helpful assistant that employs a neurosymbolic method. Given the following context and question, please follow these steps:
1. Extract all relevant facts from the context and represent them as symbolic facts using the format `FACT(entity, attribute, value)`.
2. Generate executable Python code that uses the extracted symbolic facts to compute the final answer.
3. Finally, output only the final answer.
#CONTEXT
{text}
#ENDCONTEXT
#QUESTION
What is the largest special magic number?

> **NSAR+3 Prompt Template**
>
> You are a helpful assistant that employs a neurosymbolic method combining chain-of-thought, ReAct, and self-reflection. Given the following context and question, please follow these steps:
> 1. Extract all relevant facts from the context and represent them as symbolic facts using the format `FACT(entity, attribute, value)`.
> 2. Provide a detailed, step-by-step chain-of-thought explanation of your reasoning.
> 3. Describe the action you would take (e.g., generating and executing Python code) to compute the answer.
> 4. Generate executable Python code that uses the extracted symbolic facts to compute the final answer.
> 5. Reflect on your reasoning process to verify its soundness.
> 6. Finally, output only the final answer.
> #CONTEXT
> {text}
> #ENDCONTEXT
> #QUESTION
> What is the largest special magic number?

Each of these templates is tailored to evaluate different aspects of the LLM's reasoning and retrieval capabilities. The **Vanilla** prompts test basic and multi-target retrieval scenarios, while Chain-of-Thought, ReAct, and Self-Reflection aim to improve intermediate reasoning steps. Meanwhile, **NSAR** introduces an explicit neurosymbolic approach for verifiable multi-target logic, and **NSAR+3** extends this by integrating advanced prompting strategies for even more robust reasoning.

