# OpenReview forum: "Enhancing Large Language Models with Neurosymbolic Reasoning for Multilingual Tasks"
_nesyconf.org/NeSy/2025/Conference — NeSy 2025 Poster_

### Official Review · Reviewer_1p4P · 2025-04-02
**Accept, with a suggestion of small change in narrative, and further details on results**

**Rating:** 7
**Confidence:** 5

**Review:**

The paper presents the NeuroSymbolic Augmented Reasoning (NASR) framework, that allows for multilingual retreival from long context for a symbolic-oriented large-language-model (LLM) prompt engineering. NSAR extracts symbolic facts from text and generates Python code using two LLMs, GPT-4o-mini and Lllama-3.2 90b. The authors evaluate NSAR across seven languages with varying context lengths and differing script-types to identify "needles in a haystack" and answer a query regarding the maximum of three "magic numbers" existing in the given context. They show that NSAR outperforms both vanilla RAG baselines, as well as other promting strategies: Chain-of-Thought, ReAct, and Self-Reflection.

The paper is well written, and mostly clear in its contribution to the field of Neurosymbolic AI. The authors have clearly made a series of structured experiments and discuss their results thoroughly in the paper. Particularly important are the comparison of results of various promting strategies on a hard-to-solve dataset, as well as identifying the distribution of failure cases across the different stages of inference in the NSAR pipeline (i.e. Retrieval failures, fact extraction failures, code generation failures).

However, there are a few issues that I find with the paper as written.

1. It is hard to interpret how the failure cases progress throughout out the stages of the inference pipeline. The comparison of retreival failures vs LLM failure is clear, as well as the comparison of the two LLM failure types (Fact extraction vs. Code generation), but it is unclear how all three play a role or are distributed throught the total failure cases. In addition, in Figure 7, the y-axis potrays "Failures (%)" but it is unclear what those failures are a percentage of. Are they a percentage of total queries, or a percentage of all failures (which would mean that only 4-5% of failures are LLM failures, which does not match what is displayed in Figure 3(a).

2. It would seem that the NSAR framework should be considered more of a LLM prompting strategy, rather than a neurosymbolic framework, as declared in the contribution section of this paper. This is not to diminish the contribution, but to clarify the specific contribution that leads to improved accuracy on the presented dataset task ("three needles"). The NSAR promt template provided on the bottom of page 4 suggests that it relies on prompt engineering to extract facts and generate task, relying in other words on the non-neurosymbolic operation of a LLM. The ability to enforce a LLM to think extract symbolic information, and then utilise those facts in a python code script has been shown in this work to increase accuracy, which is an important contribution. In order to check whether the NSAR approach truly goes beyond the other promting strategies, experimentation would need to be conducted by ablating the retreival component.

3. Following on point 2, it is unclear whether the results of accuracy from the other prompting strategies are from applying those stratgies with the context retrieved by the RAG system, or not. It is assumed that it is, and therefore the comparison is valid, but this should be declared definitively.

4. There seems a lack of discussion or analysis on why GPT-4o-mini performance differs from Llama-3.2. Does llama work well with the NSAR approach due to its larger context window, or larger parameters size? Or is there some other component that lead to NSAR showing different improvements with each model, over the performance of the baselines.

I recommend accepting this paper, with minor revisions, for Nesy 2025. A small change in narrative will make this paper stronger and will dissipate any hard questions from the Neurosymbolic AI community.

**Anonymity:**

Disclose identity

---

### Official Review · Reviewer_b4PC · 2025-04-03
**The authors present NSAR, a method that integrates symbolic extraction with LLM-based inference by having the model generate Python code for final verification. Their experiments cover seven languages and very long contexts. The approach seems methodologically sound and offers an interesting step beyond standard retrieval-augmented generation. However, my main concerns are about the overreliance on powerful models (90B and GPT-4o mini), loose definitions of “neurosymbolic” and "reasoning", and an inadequate analysis of code-generation failure modes.**

**Rating:** 6
**Confidence:** 4

**Review:**

Strengths:
1. A rigorous evaluation setup: The authors systematically test across multiple context lengths and k-values.
2. A detailed experimentation description: Evaluating on seven languages and the use of their “needle in a haystack” design to effectively test multi-target retrieval.
3. Practical transparency: Extracting structured facts and generating Python code, they describe an interpretable reasoning pipeline. Shared prompt templates and needle examples aid in reproducibility.
4. Comparison with relevant prompting techniques: Chain-of-Thought, ReAct, and Self-Reflection baselines highlight where purely neural methods are lacking and how explicit symbolic steps can help.



Weaknesses:
1. Overreliance on powerful language models: Most experiments use very powerful LLMs (GPT-4o-mini and Llama 3.2 with 90B parameters). Testing smaller or more open-source models would clarify how much success is due to scale rather than the neurosymbolic design.
2. Loose terminology: The paper’s use of “neurosymbolic” and “reasoning” feels broad. Code generation here is helpful but relatively simple. More explicit symbolic logic or constraints could better justify the neurosymbolic label.
3. Limited error analysis for code generation: While code failures are mentioned, there is little depth in analyzing patterns of errors. This is a critical task and warrants investigation into why and how the failure modes emerged.
4. Scope of symbolic reasoning: The symbolic extraction mostly handles numeric comparisons or direct lookups. Future work could expand into more complex symbolic domains to test generalizability. Additionally, the scope of investigation seems overly narrow, and future work can expand on using "reasoning" models to help with more generalizable tasks.

**Anonymity:**

Remain anonymous

---

### Official Review · Reviewer_jXTK · 2025-04-04
**Enhancing Large Language Models with Neurosymbolic Reasoning for Multilingual Tasks**

**Rating:** 8
**Confidence:** 4

**Review:**

The paper “Enhancing Large Language Models with Neurosymbolic Reasoning for Multilingual Tasks” proposes a neurosymbolic reasoning module (NSAR) to model multi-target reasoning in cross-lingual settings. The authors compare their NSAR method with classical reasoning approaches like retrieval-augmented generation, chain-of-thought, Reasoning and Acting (ReAct) or Self-Reflection. The results show a tendency that the NSAR approach outperforms alternative reasoning methods in large language models.

The paper is carefully written and does not contain significant formal or stylistic errors. The appendix contains additional information about the “needle templates” and prompt templates used in the paper. The figures support the claims of the text and are reasonably used in the paper. The formal aspects of the paper are ok as far as I can see this.

The presented approach sounds interesting to me. Enhancing reasoning abilities of LLMs by chain-of-though, ReAct, or Self-Reflection are classical attempts to gain better and explainable reasoning abilities of LLMs. NSAR seems to outperform these methods by extracting relevant sentences of the potentially large context first and then generate Python code for answering the question. The advantage is an explicit and executable representation of knowledge. From an XAI perspective this seems to be a very transparent and reasonable approach.

Unfortunately, the NSAR steps of symbolic fact extraction and Python Code Generation are rather broadly described without giving a lot of details. If I understand this correctly, the extracted facts have some similarities with RDF triples. This snippets of knowledge seem to be appropriate for specifying some relevant aspects of knowledge items from the input text. Nevertheless, the limitations of such an approach are obvious, because the resulting representation strength of these triples is rather limited. Furthermore, logical reasoning based on such facts is similarly restricted. The authors should more clearly specify the appropriate limitations of such an approach concerning facts of the input text.

I am not sure whether the 3-needles test in a cross-lingual setting is sufficient for a thorough evaluation of the system. The reader would like to know to which extent the proposed architecture can address other reasoning tasks like ANLI or e-SNLI. Furthermore, due to the broad and popular recognition of DeepSeek and its performance when applied to large-scale reasoning tasks like MMLU, GPQA etc.: would NSAR help in improving the performance of such classical reasoning tasks?

In total, I think the paper addresses an important aspect of neurosymbolic reasoning and learning. The paper is well written and contains an interesting architecture for integrating structured knowledge into a neural system.

**Anonymity:**

Remain anonymous